# Socioeconomic Inequalities in Metabolic Syndrome by Age and Gender in a Spanish Working Population

**DOI:** 10.3390/ijerph181910333

**Published:** 2021-09-30

**Authors:** Manuela Abbate, Jordi Pericas, Aina M. Yañez, Angel A. López-González, Joan De Pedro-Gómez, Antoni Aguilo, José M. Morales-Asencio, Miquel Bennasar-Veny

**Affiliations:** 1Research Group on Global Health, University of the Balearic Islands, 07122 Palma, Spain; manuela.abbate@uib.es; 2Health Research Institute of the Balearic Islands (IdISBa), 07120 Palma, Spain; jordi.pericas@uib.es (J.P.); depedro@uib.es (J.D.P.-G.); aaguilo@uib.es (A.A.); miquel.bennasar@uib.es (M.B.-V.); 3Nursing and Physiotherapy Department, University of the Balearic Islands, 07122 Palma, Spain; 4School of Odontology ADEMA, University of the Balearic Islands, 07009 Palma, Spain; angarturo@gmail.com; 5Prevention of Occupational Risks in Health Services, Balearic Islands Health Service, 07003 Palma, Spain; 6Department of Nursing, Universidad de Málaga, 29071 Málaga, Spain; jmmasen@uma.es; 7Instituto de Investigación Biomédica de Málaga (IBIMA), 29010 Málaga, Spain; 8CIBER de Epidemiología y Salud Pública (CIBERESP), 28029 Madrid, Spain

**Keywords:** socioeconomic status, socioeconomic status gradient, metabolic syndrome, prevalence of metabolic syndrome

## Abstract

Lower socio-economic status (SES) is significantly associated with metabolic syndrome (MS) prevalence, possibly affecting women more than men, although evidence in Spain is still limited. The present cross-sectional study analyzed the association between MS and SES by age and gender among 42,146 working adults living in the Balearic Islands (Spain). Prevalence was higher in men (9.4% by ATP-III; 12.3% by IDF) than women (3.8% by ATP-III; 5.7% by IDF) and in the lower social class (7.9% by ATP-III; 10.7% by IDF) than the higher (4.1% by ATP-III; 5.9% by IDF). The SES gradient in MS prevalence was larger in women (PR 95% CI: 3.38, 2.50–4.58 by ATP-III; 3.06, 2.43–3.86 by IDF) than in men (1.23, 1.06–1.41 by ATP-III; 1.15, 1.03–1.30 by IDF) and was already evident from early adulthood, reaching the highest ratio at the late stages of middle adulthood (4.34, 1.11–16.98). Among men, it was significant during the late stages of early adulthood only (1.80, 1.19–2.73). Lower SES influenced MS prevalence in both genders, however, women seemed more affected than men. From a public health perspective, SES could be strongly associated with the burden of MS; in an effort to reduce its prevalence, public health policies should focus on gender differences in socio-economic inequality and consider women with low socio-economic resources as a priority.

## 1. Introduction

Previous evidence suggests that socio-economic inequalities as well as demographic differences are associated with an unequal distribution of health and disease in high-, middle-, and low-income countries [1,2]. The metabolic syndrome (MS), a cluster of metabolic risk factors related to cardiovascular (CV) disease and type 2 diabetes (T2D) [3], follows the same trend, as its prevalence shows an inverse association with socio-economic status (SES). Lower incomes and lower education levels are associated with a higher prevalence of MS, whilst a higher SES is a protective factor [4,5,6,7,8,9,10], possibly from early adolescence [11], although its beneficial effect may eventually diminish with increasing age [12,13]. In recent reports, MS prevalence was generally higher in men than women, however, only up to the age of 50 years, after which, such a trend was reversed. Post-menopausal women became, in fact, increasingly predisposed to developing MS and suffered from CV events more than men of the same age [14,15]. Moreover, specific to MS and CV disease, the SES gradient phenomenon seems more evident in women of any age than in men [4,16,17,18,19,20], whilst, for all other health outcomes, such as cancer, it appears stronger for men than women [20].

Specific to the Spanish population, a study carried out in adults aged 35–74 years showed a higher prevalence of MS in men up to 55 years of age and in women over the age of 65, who also presented a higher coronary risk associated with the presence of MS [21]. MS prevalence in the working population, which overall was estimated to be 10.2% by the Modified National Cholesterol Education Program Adult Treatment Panel-III (NCEP-ATP III) criteria [5], was significantly higher among men than women [4,5,22,23] and among older workers and those in the lower SES [4,5,22]. However, when looking at MS prevalence in relation to SES in men and women separately, information is very limited. An analysis of data from 259,014 participants of the Ibermutuamur Cardiovascular Risk Assessment (ICARIA) study [4] showed that prevalence of MS in females was more frequent in the lower social class as compared to the higher, whilst, in men, it was similar between the two classes. Such findings relate to those of other countries, in which women from low socio-economic status are at a higher risk than men of presenting MS [17,18,19], however, more evidence is needed.

The main aim of the present study was to analyze the association between MS and SES by age and gender in a large sample of working adults living in the Balearic Islands.

## 2. Materials and Methods

The present study is a cross-sectional analysis of data from 42,146 working adults employed in public administration, healthcare, and postal services and living in the Balearic Islands (Spain). Screening and inclusion of participants were carried out between January 2008 and December 2010 during voluntary routine occupational health visits. Inclusion criteria included: age between 18–65 years and being gainfully employed; exclusion criteria included: previously diagnosed diabetes (type 1 and 2), previously diagnosed CV disease or a history of CV event, anemia, current treatment with systemic steroids, active cancer or a history of malignancy in the previous 5 years, or pregnancy.

Importantly, eligibility criteria were defined taking into consideration the conclusions drawn by the World Health Organization (WHO) Expert Consultation held in 2009 [24], which redefined MS as a pre-morbid condition rather than a clinical diagnosis, thus suggesting the exclusion of individuals with established diabetes and CV disease.

A total of 54,236 workers eligible for study inclusion were informed about the purpose of the study and invited to participate; 43,265 agreed to participate and signed the written informed consent. Of those, 42,146 presented available data on social class and were included in the cross-sectional analysis.

The study protocol was in accordance with the Declaration of Helsinki and was approved by the Ethics Committee of Clinical Research of the Balearic Islands (CEI-IB) with reference number 1887 as well as by the Gestion Sanitaria de Mallorca (GESMA) Ethics Research Commission.

### 2.1. Study Variables

After acceptance, information on socioeconomics, a complete medical history including family and personal history, and smoking status (current, former, never) were recorded.

Data on occupation were collected in accordance with the Spanish National Classification of Occupations (CNO-2011). According to the job position declared, participants were assigned to one of the three main social classes defined by the classification of occupational social class (CSO-SEE 2012) based on the Neo-Weberian class analysis proposed by Goldthorpe and adapted for the Spanish population by the Spanish Epidemiology Society [25,26]. Class I included managerial, administrative, and professional occupations with a university degree; class II included intermediate occupations and self-employed; and class III included manual workers.

Anthropometric measurements were taken by trained staff following the recommendations of the International Standards for Anthropometric Assessment (ISAK) [27]. Specifically, body weight was measured to the nearest 0.1 kg using a mechanical column scale (Seca 700, Seca GmbH, Hamburg, Germany). Height was measured to the nearest 0.5 cm using a scale mounted telescopic stadiometer (Seca 220, Seca GmbH, Hamburg, Germany). BMI was calculated using the standard formula (weight in kg divided by squared height in m, kg/m^2^). Waist circumferences (WC) were measured using a flexible steel tape (Lufkin Executive Thinline W606, Apex Tool Group, Dallas, TX, USA) with the participant standing upright with feet together and arms hanging freely at the sides. WC was taken at midway between the last rib and the top of the iliac crest; the plane of the tape was perpendicular to the long axis of the body and parallel to the floor. Each anthropometric measurement was repeated three times, and averaged values were used for analysis.

Blood pressure was measured after a resting period of 10 min in supine position using an automatic and calibrated sphygmomanometer (OMRON M3, OMRON Healthcare Europe, Spain). Measurements were taken three times with a one-minute gap in between; the average value was used for analysis.

Venous blood samples were taken from the antecubital vein in suitable vacutainers following a 12 h overnight fast. Samples were then centrifugated to obtain serum (15 min, 1000 g, 4 °C), which was stored at −20 °C and analysed for fasting plasma glucose (FPG), total cholesterol (T-Chol), HDL-cholesterol (HDL-C), and triglycerides (TG) within 3 days by standard procedures using an autoanalyzer (SYNCHRON CX^®^ 9 PRO, Beckman Coulter, Brea, CA, USA).

The MS was defined in accordance with the International Diabetes Federation (IDF) [28] and the Modified National Cholesterol Education Program Adult Treatment Panel-III (NCEP-ATP III) [29], excluding participants with established diabetes and CV disease. According to the IDF criteria, the diagnosis of MS is made upon the concomitant presence of increased abdominal obesity (defined as a waist circumference ≥ 94 cm in Caucasian and Black males, ≥90 cm in Asian males, and ≥80 cm in females of any race) or a BMI >30 kg/m^2^ and two of the following: (1) TG levels ≥ 150 mg/dL (1.7 mmol/L), or specific drug treatment; (2) reduced HDL-C: <40 mg/dL (1.03 mmol/L) in males and <50 mg/dL (1.29 mmol/L) in females or specific drug treatment; (3) raised blood pressure (BP): systolic BP ≥ 130 or diastolic BP ≥ 85 mm Hg or drug treatment of previously diagnosed hypertension; (4) raised FPG > 100 mg/dL (5.6 mmol/L) or specific drug treatment. According to the NCEP-ATP III criteria, the diagnosis of MS is made when at least three of the following five main traits are concomitantly present: (1) increased abdominal obesity: waist circumference ≥ 102 cm in males and ≥88 cm in females; (2) TG levels ≥ 150 mg/dL (1.7 mmol/L) or specific drug treatment; (3) reduced HDL-C: <40 mg/dL (1.03 mmol/L) in males and <50 mg/dL (1.29 mmol/L) in females or specific drug treatment; (4) raised blood pressure (BP): systolic BP ≥ 130 or diastolic BP ≥ 85 mm Hg or drug treatment of previously diagnosed hypertension; (5) raised FPG ≥ 100 mg/dL (5.6 mmol/L) or specific drug treatment. The main differences between the two criteria relate to the threshold used to define abdominal obesity by waist circumference, which is lower and ethnicity-based when employing the IDF definition, and the lower limit of FPG, which in the IDF is 101 mg/dL whilst in the NCEP-ATP III is 100 mg/dL. Thus, depending on which criteria is used, prevalence of MS can vary significantly [30,31,32].

### 2.2. Statistical Analyses

Descriptive statistical analysis was carried out to summarize quantitative data variables through measures of central tendency and homogeneity, and qualitative data variables through analysis of proportions. Variables distributions were assessed by using the Kolmogorov–Smirnov test and by visual inspection of Q-Q normal probability plots. For bivariate analysis with quantitative data, Student’s *t*-test for comparison of means was used in cases of normally distributed variables (using the Levene’s test for equal variances), whilst non-parametric tests such as the Mann–Whitney U test (independent samples) were used for variables showing non-normal distribution. Mean comparison across the three social classes was carried out by using ANOVA. Analysis of qualitative variables was carried out by using the Chi-squared test.

Prevalence ratios (PR) and corresponding 95% confidence intervals (CI) were calculated to compare the frequency of MS among different social classes. Significance was accepted at *p* < 0.05.

Statistical analysis was carried out using IBM SPSS Statistics 20.0 software (SPSS/IBM, Chicago, IL, USA).

## 3. Results

Of the 42,146 participants included in the analysis, 23,956 (56.8%) were males and 18,190 (43.2%) were females. Mean (standard deviation, SD) age was 39.25 (10.29), BMI was 25.79 (4.49) kg/m^2^, systolic BP was 120.36 (15.91) mmHg, diastolic BP was 73.22 (10.78) mmHg, FPG was 85.47 (11.75) mg/dL, T-Chol was 192.14 (37.11) mg/dL, HDL-C was 51.80 (8.26) mg/dL, TG was 104.20 (72.73) mg/dL, and waist circumference was 82.14 (11.43) cm. All subjects were Caucasians.

Obesity was present in 6708 (15.9%) subjects, while 15,060 (35.7%) were overweight, 19,657 (46.6%) had a normal BMI, and 721 (1.7%) were underweight. As for smoking habits, 14,965 (35.5%) were current smokers, 19,970 (47.4%) non-smokers, and 7211 (17.1%) ex-smokers. The percentage of prevalent MS by ATP III criteria was 7.0% (*n* = 2954) whilst, by IDF criteria, it was 9.5% (*n* = 3993). The distribution of the whole sample by social class was as follows: 5231 (12.4%) subjects belonged to class I, 10,523 (25.0%) to class II, and 26,392 (62.6%) to class III.

General characteristics of the study population by gender are displayed in Table 1. In general, men were more likely to belong to social class III, smoke, be overweight or obese, and present higher levels of systolic and diastolic BP, T-Chol, FPG, TG, and WC and lower levels of HDL-C as compared to women. The prevalence of MS by both criteria was significantly higher in men than in women.

General characteristics of the study population stratified by social class are displayed in Table 2. Across the three classes, from I to III, age, BMI, systolic and diastolic BP, TG, and WC increased significantly, whilst HDL-C decreased. T-Chol and FPG were significantly higher in classes II and III as compared to class I. Subjects in social classes II and III were more likely to be overweight and obese than those in social class I, which were mostly normal-weight; accordingly, prevalence of MS was increased across classes from class I to class III. Lastly, the proportion of active smokers was higher in social class III than in the other two classes; in men, the percentages of smokers were 27.2% in social class I, 26.1% in social class II, and 41.5% in social class III (*p* < 0.0001), with a PR between classes I and III of 1.52 (95% CI 1.42 to 1.63). In women, the percentages of smokers were 26.9% in class I, 29.0% in class II, and 38.2% in class III (*p* < 0.0001), with a PR between extreme classes of 1.44 (95% CI 1.36 to 1.53) (data not shown).

Weight statuses stratified by social class in men and women are displayed in Table 3. For both men and women, the analysis showed higher PR between social class I and III for obesity, with a higher prevalence among those subjects belonging to the lower social class. When considering overweight status, women in social class III presented a higher prevalence as compared to women in social class I. For men, on the other hand, the trend was reversed, as those in social class III were less likely to be overweight compared to men in social class I.

Prevalence of MS assessed by both the ATP-III and the IDF criteria stratified by social class in men and women is displayed in Table 4. In men, prevalence of MS by ATP-III criteria was 8.01% in social class I, 8.72% in social class II, and 9.82% in social class III (*p* = 0.004). In women, MS by ATP-III criteria was 1.35% in social class I, 3.85% in social class II, and 4.6% in social class III (*p* < 0.001). When MS was assessed by IDF criteria, its prevalence in men was 10.97% for class I, 11.56% in class II, and 12.68% in class III (*p* = 0.016), whilst in women, prevalence was 2.30% for class I, 5.39% in class II, and 7.05% in class III (*p* > 0.001). In both cases, the PR between extreme classes showed a clear higher prevalence in the lower social classes, this difference being higher in women.

Prevalence ratios of presenting MS between extreme classes by age groups in men and women are displayed in Table 5. Men in social class III aged 35–39 years had a significantly higher prevalence of MS by both ATP-III and IDF criteria than men in social class I. Women in social class III aged 30–34, 40–44, 45–49, and 55–59 years had a significantly higher prevalence of MS by ATP-III criteria than women in social class I. When considering prevalence of MS by IDF criteria, women belonging to social class III aged 25–29, 30–34, 35–39, 40–44, 45–49, and 55–59 presented a significantly higher prevalence than women of the same age groups belonging to social class I.

BMI, T-Chol, and MS components such as systolic and diastolic BP, HDL-C, TG, FPG, and WC stratified by social class and age groups for men and women are displayed in Figure 1. Across all ages, men presented higher BMI values than women. In men, BMI increased with age and was generally higher in social class III than in the other two classes. Women’s BMI was consistently higher in social class III than in the other two classes for all age groups.

Both systolic and diastolic BP were higher in men than women across all ages and social classes and increased with age in both genders. Women in social classes II and III tended to present higher BP levels than women in social class I, especially after the age of 50 years, whilst differences in BP between social classes in men were smaller.

Starting at similar initial levels, HDL-C decreased with age in both genders and across the three social classes. It is noteworthy that women in social class I presented higher HDL-C levels than women in the other two classes and maintained such difference across all age ranges.

Levels of TG were similar between men and women only up to the age of 25, after which, in men, levels tended to increase gradually with age independently of social class. In women, TG levels tended to remain stable between 20 and 50 years and then progressively increased in social class III more than in social class I.

Across all age ranges and social classes, men presented significantly higher FPG levels than women. No appreciable differences were found between social classes for neither men nor women.

Finally, across the three social classes, WC significantly increased with age in both men and women.

## 4. Discussion

The present cross-sectional study shows that, in the working population of the Balearic Islands, overall prevalence of MS was 7% by ATP III and 9.5% by IDF criteria. Prevalence was higher in men (9.4% by ATP-III; 12.3% by IDF) than women (3.8% by ATP-III; 5.7% by IDF), and, accordingly, men were more overweight and obese and presented a worse blood lipid profile and higher levels of FPG and BP than women. Prevalence of MS was higher in the lower social class (7.9% by ATP-III; 10.7% by IDF) than in the higher (4.1% by ATP-III; 5.9% by IDF). As compared to subjects in social class I, those in social class III were more likely to be men, older, and present a worse cardiometabolic profile, including higher rates of overweight and obesity.

When looking at the effect of SES on MS prevalence for men and women separately, both genders were affected by social class differences; however, while the increase of MS prevalence across social classes in men was gradual, the SES gradient in women was particularly evident, as MS prevalence in class III was three-fold greater than in social class I. Moreover, among women, such difference was already noticeable from early adulthood, reaching the widest gap at the late stages of middle adulthood. On the other hand, social class differences in MS prevalence among men were significant during the late stages of early adulthood only and, in any case, smaller than those among women.

Those components of the MS that seemed to be responsible for social class differences in women were BP, HDL-C, and TG, especially after the age of 50 years, while waist circumference and BMI differences appeared from much earlier on. Conversely, men in social class III presented higher levels of FPG and TG and lower levels of HDL-C than men in social class I, especially at around the late stages of early adulthood.

The prevalence of MS, as defined by ATP-III criteria, observed in the present analysis is similar to that observed in previously published Spanish studies on the adult working population [4,5,22]. Similarly to ours, these studies also observed that the prevalence of MS was higher in men than women, increased with age, and was greater in the lower social classes as compared to the higher [4,5,22].

Numerous European studies carried out on the general population, not exclusively adult workers, also coincide in these aspects. Specific to the association between MS and gender, what is generally observed is a higher prevalence of MS in men than in women, regardless of their socioeconomic status [33,34,35,36], although, in a large Norwegian study, a similar prevalence was observed in both genders [37]. Increasing age has also been strongly associated with the prevalence of MS and a worse prognosis in both men and women [9,37,38]. Lastly, with respect to SES and MS, throughout Europe, there are evident inequalities in the prevalence of MS by occupational status and educational level, as economic and social vulnerability have been observed to be a significant determinant of MS [9,33,34,36,39,40].

Gender differences in the prevalence of MS in response to SES have been seldom investigated, although the association tends to be stronger in women than in men. Specific to Spain, the ICARIA study, a cross-sectional analysis of the working population observed that the social gradient in the prevalence of MS was significant in women only, as men showed no differences in prevalence between social classes [4]. Likewise, in a Portuguese study including middle-aged and old adults [12], belonging to a lower social class was associated with MS prevalence in females but not in males. In our study, both sexes were affected by social class, however, the association was stronger in women than in men. Similar results were observed in the Whitehall II Study [39], which concluded that the effect of income category was stronger in women than in men; additionally, MS prevalence increased consistently across the six income categories among men, while, for women, a higher prevalence was only found in the three lowest categories.

The effect of socio-economic inequality on MS prevalence in women seems to operate significantly and independently from the early stages of life [40,41]. It has been observed that women born in lower socio-economic conditions were less likely than men to have a university degree or ascend to the highest social class when adults; their permanence into a disadvantaged socio-economic condition from childhood possibly exposed them to psychosocial hazards and material deprivation, influencing their health choices and behaviors from an early age, ultimately determining their health outcomes as adults [40,41,42]. According to our results, the socio-economic gap among women influenced MS prevalence from early on, however, its strongest effect was exerted during the late stages of middle adulthood. What seemed to affect women in the lower social class more than in the higher ones were waist circumference and BMI, then, from the age of 50 years, BP, HDL-C, and TG levels. Similar to our results, also in the Whitehall II Study, it was observed that a high waist-to-hip ratio, low HDL-C, and high TG but not BP were inversely related to income categories in women [39], however, no distinction was made between younger and older populations.

Previous studies observed a surge in MS prevalence among post-menopausal women [14,15], nevertheless, in our sample, we also observed that women who reached the average age for menopause seemed to be more sensitive to the SES gradient phenomenon. Supporting evidence, although limited, comes from longitudinal observational studies which showed that the risk of MS during menopause was associated with low SES [43,44,45,46] and specifically with lower education attainment [43] and unemployment [44] besides lifestyle factors such as exercise and alcohol consumption [45] and, possibly, with the diminishing capacity that women in low SES environments had to recover, over the years, from chronic exposure to stressors, which causes negative emotional experiences [46].

Preventative efforts to tackle the effect that SES has on MS should not be limited to targeting health behaviors but should also reduce this effect. Identifying factors responsible for the SES gradient phenomenon, which can be used as targets for intervention, is of foremost importance.

The main strength of the present study is that is representative of the study population. The size of the study sample was 42,146 subjects (56.8% men and 43.2% women), which represents 9.45% of the total active working population of the Balearic Islands in 2011. Moreover, the percentage of men and women in the sample is similar to that observed for the entire workforce in the Balearic Islands (54.29% men and 45.71% women) [47]. Limitations include the cross-sectional nature of the study, which only allows the description of associations, and the voluntary nature of the occupational health control, which could establish a bias in the selection process. It is, in fact, unknown whether the inclusion of workers who did not attend the health visit and who were possibly healthier would modify the prevalence of MS.

## 5. Conclusions

Our results indicate that lower socio-economic status significantly influenced MS prevalence in both genders, however, the association was stronger in women than in men. Among women, the SES gradient in MS prevalence was already evident from early adulthood, although the highest ratio was reached at the late stages of middle adulthood, or, rather, at around the age of menopause. In men, social class differences in MS prevalence were significant during the late stages of early adulthood only.

## Figures and Tables

**Figure 1 ijerph-18-10333-f001:**
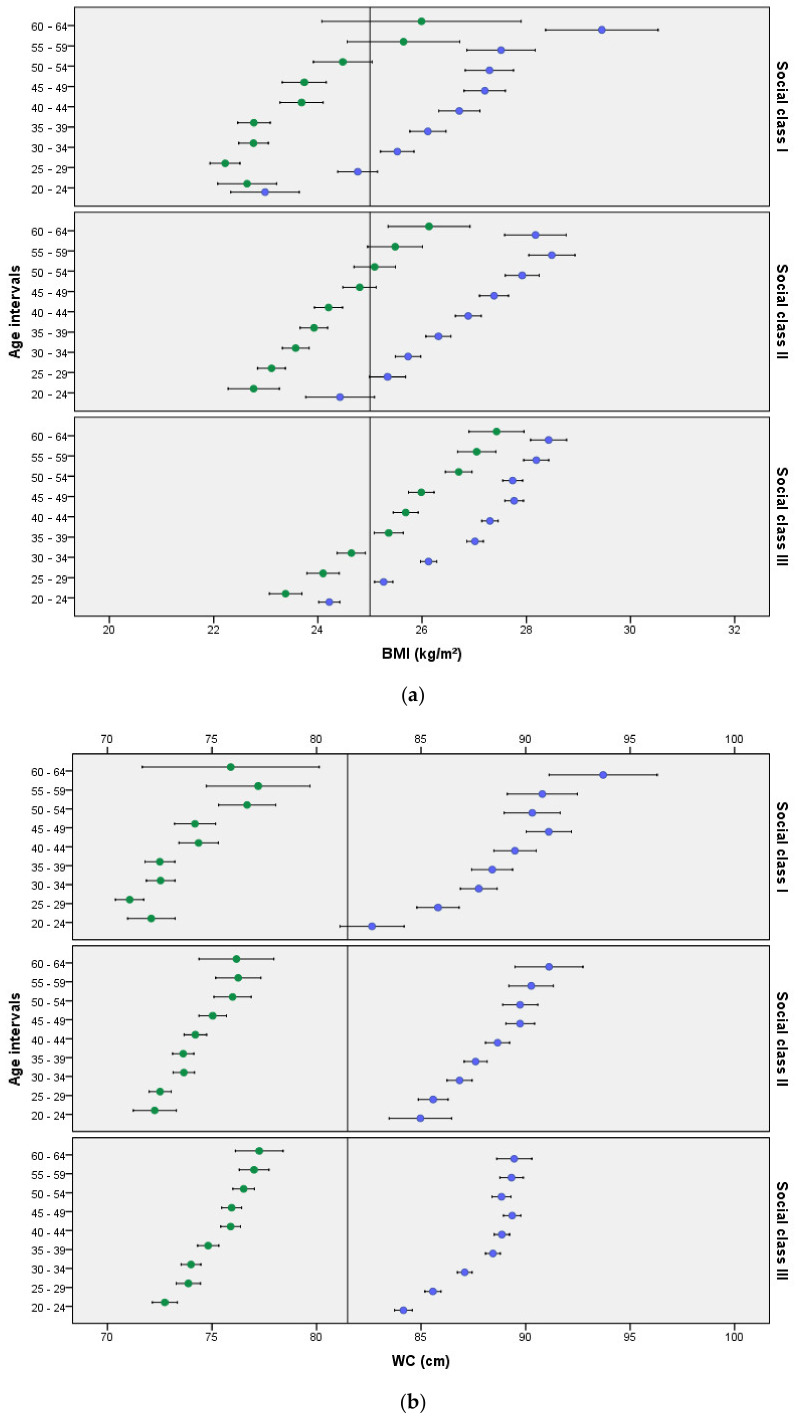
Distribution of mean and 95% CI of BMI, systolic and diastolic BP, T-Chol, HDL-C, TG, glucose, and WC by age and social class in men and women for different age groups. Panel description: (**a**) BMI: body mass index; (**b**) WC: waist circumference; (**c**) T-Chol: total cholesterol; (**d**) HDL-C: high density lipoprotein cholesterol; (**e**) TG: triglycerides; (**f**) Diastolic BP: diastolic blood pressure; (**g**) Systolic BP: systolic blood pressure; (**h**) FPG: fasting plasma glucose.

**Table 1 ijerph-18-10333-t001:** General characteristics of study sample stratified by gender.

	Men	Women	*p*
*n* (%)	23,956 (56.8)	18,190 (43.2)	
Age (y)	39.54 (10.39)	38.87 (10.13)	<0.001
BMI (kg/m^2^)	26.68 (4.13)	24.60 (4.68)	<0.001
Systolic BP (mmHg)	125.20 (15.18)	113.99 (14.52)	<0.001
Diastolic BP (mmHg)	75.61 (10.68)	70.08 (10.09)	<0.001
T-Chol (mg/dL)	193.96 (38.12)	189.75 (35.60)	<0.001
HDL-C (mg/dL)	49.98 (6.99)	54.21 (9.13)	<0.001
FPG (mg/dL)	87.34 (12.13)	83.01 (10.74)	<0.001
TG (mg/dL)	120.33 (85.59)	82.97 (42.59)	<0.001
WC (cm)	7.97 (9.41)	74.45 (9.06)	<0.001
Social class (*n* (%))			<0.001
I	2195 (9.2)	3036 (16.7)	
II	4849 (20.2)	5674 (31.3)	
III	16,912 (70.6)	9480 (42.1)	
Weight status (*n* (%))			<0.001
Underweight	155 (0.6)	588 (3.1)	
Normal weight	8097 (36.8)	11,185 (59.9)	
Overweight	10,628 (43.9)	4633 (24.8)	
Obesity	4521 (18.7)	2275 (12.2)	
Smoking habits (*n* (%))			<0.001
Current smokers	8972 (37.1)	6235 (33.4)	
Non-smokers	10,536 (43.5)	9935 (53.2)	
Ex-smokers	4703 (19.4)	2511 (13.4)	
MS prevalence (*n* (%))			<0.001
ATP-III	2261 (9.4)	693 (3.8)	
IDF	2948 (12.3)	1045 (5.7)	

Data are expressed as mean (standard deviation) and counts (percentage). *p*-values of quantitative data were determined by nonpaired Student’s *t*-test or Mann–Whitney U according to variables distribution. *p*-values of qualitative data were determined by Chi-squared test. Abbreviations: ATP-III: Adult Treatment Panel-III; BP: blood pressure; BMI: body mass index; FPG: fasting plasma glucose; HDL-C: high density lipoprotein cholesterol; IDF: International Diabetes Federation; MS: metabolic syndrome; T-Chol: total cholesterol; TG: triglycerides; WC: waist circumference.

**Table 2 ijerph-18-10333-t002:** General characteristics of study sample stratified by social class.

	I	II	III	*p*
*n* (%)	5231 (12.4)	10,523 (25.0)	26,392 (62.6)	
Gender (*n* (%))				<0.001
Men	2195 (42.0)	4849 (46.1)	16,912 (64.1)	
Women	3036 (58.0)	5674 (53.9)	9480 (35.9)	
Age (y)	37.19 (9.48)	39.27 (9.45)	39.65 (10.70)	<0.001
BMI (kg/m^2^)	24.38 (4.12)	25.24 (4.31)	26.29 (4.56)	<0.001
Systolic BP (mmHg)	115.45 (14.82)	118.64 (15.27)	122.02 (16.08)	<0.001
Diastolic BP (mmHg)	71.43 (10.18)	72.43 (10.7)	73.89 (10.88)	<0.001
T-Chol (mg/dL)	188.23 (34.69)	193.01 (35.88)	192.58 (38.01)	<0.001
HDL-C (mg/dL)	54.43 (10.50)	52.04 (8.20)	51.19 (7.66)	<0.001
FPG (mg/dL)	84.77 (10.39)	84.77 (11.09)	85.89 (12.23)	<0.001
TG (mg/dL)	93.96 (58.39)	99.43 (65.17)	108.13 (77.67)	<0.001
WC (cm)	79.57 (12.09)	80.53 (11.28)	83.29 (11.19)	<0.001
Weight status (*n* (%))				<0.001
Underweight	170 (3.2)	191 (1.8)	360 (1.4)	
Normal weight	3061 (58.5)	5494 (52.2)	11,102 (42.1)	
Overweight	1527 (29.2)	3490 (33.2)	10,043 (38.1)	
Obesity	473 (9.0)	1348 (12.8)	4887 (18.5)	
Smoking habits (*n* (%))				<0.001
Current smokers	1404 (26.8)	2912 (27.7)	10,649 (40.3)	
Non-smokers	3214 (61.4)	5593 (53.2)	11,163 (42.3)	
Ex-smokers	613 (11.7)	2018 (19.2)	4580 (17.4)	
MS prevalence (*n* (%))				<0.001
ATP-III	217 (4.1)	642 (6.1)	2095 (7.9)	
IDF	311 (5.9)	867 (8.2)	2815 (10.7)	

Data are expressed as mean (standard deviation) and counts (percentage). *p*-values of quantitative data were determined by ANOVA test. *p*-values of qualitative data were determined by Chi-squared test. Abbreviations: ATP-III: Adult Treatment Panel-III; BP: blood pressure; FPG: fasting plasma glucose; HDL-C: high density lipoprotein cholesterol; IDF: International Diabetes Federation; MS: metabolic syndrome; T-Chol: total cholesterol; TG: triglycerides; WC: waist circumference.

**Table 3 ijerph-18-10333-t003:** Weight status stratified by social class in men and women.

	Social Class			Total	*p*	PR (95% CI) *
	I	II	III			
Men (*n* (%))	2195 (9.7)	4849 (20.2)	16,912 (71.1)	23,956		
Weight status (*n* (%))						
Underweight	3 (0.1)	13 (0.3)	135 (0.8)	151 (0.6)	<0.001	5.84 (1.89 to 18.1)
Normal weight	885 (40.3)	1765 (36.4)	6143 (36.3)	8793 (36.7)	<0.001	0.90 (0.86 to 0.95)
Overweight	1010 (46.0)	2231 (46.0)	7290 (43.1)	10,531 (44.0)	<0.001	0.94 (0.90 to 0.98)
Obesity	297 (13.5)	840 (17.3)	3344 (19.8)	4481 (18.7)	<0.001	1.46 (1.31 to 1.62)
Women (*n* (%))	3036 (16.7)	5674 (31.2)	9480 (52.1)	18,190		
Weight status (*n* (%))						
Underweight	167 (5.5)	178 (3.1)	225 (2.4)	570 (3.1)	<0.001	0.43 (0.37 to 0.5)
Normal weight	2176 (71.7)	3729 (65.7)	4959 (52.3)	10,864 (59.7)	<0.001	0.73 (0.71 to 0.75)
Overweight	517 (17.0)	1259 (22.2)	2753 (29.0)	4529 (24.9)	<0.001	1.71 (1.58 to 1.84)
Obesity	176 (5.8)	508 (9.0)	1543 (16.3)	2227 (12.2)	<0.001	2.81 (2.43 to 3.24)

Data are expressed as counts (percentage). * Prevalence ratio between class I and III with 95% confidence interval.

**Table 4 ijerph-18-10333-t004:** Metabolic syndrome prevalence according to the ATP-III and the IDF criteria, stratified by social class in men and women.

	Social Class			Total	*p*	PR (95% CI) *
	I	II	III			
Men (*n* (%))	2195 (9.2)	4849 (20.2)	16,912 (70.6)	23,956		
MS prevalence (*n* (%))						
ATP-III	176 (8.0)	423 (8.7)	1662 (9.8)	2261 (9.4)	0.004	1.23 (1.06 to 1.41)
IDF	241 (11.0)	561 (11.6)	2146 (12.7)	2948 (12.3)	0.016	1.15 (1.03 to 1.30)
Women (*n* (%))	3036 (16.7)	5674 (31.2)	9480 (52.1)	18,190		
MS prevalence (*n* (%))						
ATP-III	41 (1.4)	219 (3.9)	433 (4.6)	693 (3.8)	<0.001	3.38 (2.50 to 4.58)
IDF	70 (2.3)	306 (5.4)	669 (7.1)	1045 (5.7)	<0.001	3.06 (2.43 to 3.86)

Data are expressed as counts (percentage). * Prevalence ratio between class I and III with 95% confidence interval. Abbreviations: ATP-III: Adult Treatment Panel-III; IDF: International Diabetes Federation; MS: metabolic syndrome.

**Table 5 ijerph-18-10333-t005:** PR (95% CI) of MS (ATP-III and IDF criteria) between extreme classes by age groups in men and women.

Age Groups (y)	*n*	MS by ATP-III Criteria		MS by IDF Criteria	
		Men	Women	Men	Women
20–24	2987	1.60 (0.23 to 11.23)	0.28 (0.04 to 1.98)	2.29 (0.33 to 16.05)	0.35 (0.13 to 0.92)
25–29	5269	1.59 (0.72 to 3.50)	2.78 (0.90 to 8.59)	1.61 (0.81 to 3.19)	2.56 (1.07 to 6.14)
30–34	6913	1.35 (0.87 to 2.09)	3.55 (1.15 to 10.98)	1.44 (0.97 to 2.15)	2.75 (1.32 to 5.75)
35–39	7124	1.80 (1.19 to 2.73)	1.77 (0.93 to 3.38)	1.41 (1.02 to 1.96)	2.39 (1.33 to 4.29)
40–44	6504	1.25 (0.90 to 1.75)	2.83 (1.36 to 5.90)	1.31 (0.98 to 1.76)	2.66 (1.52 to 4.64)
45–49	5578	1.10 (0.83 to 1.47)	2.25 (1.02 to 4.96)	1.07 (0.84 to 1.37)	2.18 (1.19 to 4.01)
50–54	4175	0.94 (0.68 to 1.30)	1.65 (0.87 to 3.13)	0.94 (0.71 to 1.25)	1.46 (0.90 to 2.37)
55–59	2525	1.29 (0.82 to 2.04)	4.34 (1.11 to 16.98)	1.08 (0.76 to 1.53)	4.02 (1.33 to 12.13)
60–64	1071	0.77 (0.48 to 1.24)	3.12 (0.46 to 21.23)	0.63 (0.44 to 0.88)	1.54 (0.54 to 4.45)

Prevalence ratio between class I and III with 95% confidence interval. Abbreviations: ATP-III: Adult Treatment Panel-III; IDF: International Diabetes Federation; MS: metabolic syndrome.

## Data Availability

Data presented in this study are available on request from the corresponding author.

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
