# Peer review of "Socioeconomic Inequalities in Metabolic Syndrome by Age and Gender in a Spanish Working Population"

_ijerph, 2021, doi:10.3390/ijerph181910333_

Round 1

Reviewer 1 Report

In the present paper, M. Abbate and coworkers analyzed the association between Metabolic Syndrome (MS) and lower Socio-economic Status (SES) by age and gender among 42,146 working adults living in the Balearic Islands (Spain). The authors concluded that lower SES influenced MS prevalence in both genders, however women seemed more affected than men. Specifically, among women, the SES gradient in MS prevalence was already evident from early adulthood, reaching the highest ratio at around the age of menopause; among men it was significant at the age of 35-39 years. Overall, I think that the paper is nice, and it could be of interest for readers and researchers, in general.

I make some suggestions for further improve the quality of the manuscript.

The authors, if possible, should incorporate in tables the dietary pattern of the patients/working adults included in the present study (e. Mediterranean-style diet, Plants-based diet, Nordic dietary pattern, etc.); in this way, I feel that the readers can better understand the epidemiological data obtained in the present clinical study and their possible application to clinical practice. Indeed, a lower SES could have a role on diet of patients/consumers with MS and, consequently on results of this study. Please discuss this crucial aspect in the revised manuscript.

“Oriental diet” is particularly abundant in isoflavones; moreover, asian women generally reported low rates of a variety of physical and psychological symptoms related to menopause and these rates were much lower than in women from Western countries. This latter could be strongly related to "oriental diet" and to above mentioned phytochemicals. Please discuss this crucial aspect in the revised manuscript considering the data here reported.

Some medicines/drugs can be used to treat MS. This aspect could interfere with results here revealed. Please discuss this aspect and eventually take this factor into account in statistical analysis.

Author Response

Reviewer 1

In the present paper, M. Abbate and coworkers analyzed the association between Metabolic Syndrome (MS) and lower Socio-economic Status (SES) by age and gender among 42,146 working adults living in the Balearic Islands (Spain). The authors concluded that lower SES influenced MS prevalence in both genders, however women seemed more affected than men. Specifically, among women, the SES gradient in MS prevalence was already evident from early adulthood, reaching the highest ratio at around the age of menopause; among men it was significant at the age of 35-39 years. Overall, I think that the paper is nice, and it could be of interest for readers and researchers, in general.

I make some suggestions for further improve the quality of the manuscript.

The authors, if possible, should incorporate in tables the dietary pattern of the patients/working adults included in the present study (e. Mediterranean-style diet, Plants-based diet, Nordic dietary pattern, etc.); in this way, I feel that the readers can better understand the epidemiological data obtained in the present clinical study and their possible application to clinical practice. Indeed, a lower SES could have a role on diet of patients/consumers with MS and, consequently on results of this study. Please discuss this crucial aspect in the revised manuscript.

R: We thank the reviewer for the suggestion. Unfortunately, due to the nature of the visits (routine occupational health visits), data on specific dietary patterns were not collected. We agree that such information could give a useful insight into understanding health differences between men and women of different social classes, however we cannot answer such question.

“Oriental diet” is particularly abundant in isoflavones; moreover, Asian women generally reported low rates of a variety of physical and psychological symptoms related to menopause and these rates were much lower than in women from Western countries. This latter could be strongly related to "oriental diet" and to above mentioned phytochemicals. Please discuss this crucial aspect in the revised manuscript considering the data here reported.

R: We thank the reviewer for highlighting the importance of diet especially in women. As previously mentioned, since information on diet were not thoroughly collected, we don’t possess data to analyze the subject. Nevertheless, we are extremely thankful for pointing out such possible association which we can certainly consider for our next research question.

Some medicines/drugs can be used to treat MS. This aspect could interfere with results here revealed. Please discuss this aspect and eventually take this factor into account in statistical analysis.

R: Thanks to the reviewer’s comment we realized that the eligibility criteria were not complete. We apologize for the confusion. As now shown in the Materials and Methods section we excluded participants with type 2 diabetes and CV disease. Eligibility criteria were defined taking into consideration the conclusions drawn by the World Health Organization Expert Consultation held in 2009 [Simmons RK, et al. Diabetologia 53, 600–605 (2010)], which redefined MS as a pre-morbid condition rather than a clinical diagnosis, thus suggesting the exclusion of individuals with established diabetes and CV disease.

Hence, none of the participants was taking treatment for MS. Of the 42,146 participants, 2,113 were taking blood pressure lowering medications, and 1,044 were taking blood lipid lowering medications. When screening for participants with MS, treatments were taken into consideration and participants with normal blood pressure or normal triglycerides levels taking either blood pressure or blood lipid lowering medications were considered as having MS if presenting the criteria defined by either the IDF or the ATP-III, as already explained the Materials and Methods section.

Reviewer 2 Report

 This is a research paper aiming to understand the health disparity in metabolic syndrome by SES and gender among 42,146 working adults employed in the public administration, health care, and postal services in Spain. Overall, this is an interesting study that provides a great overview on how gender and SES affects metabolic syndrome at the different stage by gender among working adults. It would be nice if the study can be expanded to include all adults so that more women (for example house wives) can be included and accounted for. The manuscript is well-written with the following suggestions that need to be improved or addressed:

  • Conclusion in the abstract is repeating the results. Please conclude by summarizing overall study and impact in public health.
  • The participant number is different from the number in the methods section.
  • Can the authors please provide footnotes on how the statistics were done in each table in order for the audience to understand the comparison and tests being done?
  • Typo in line 185.
  • What were the numbers in the parentheses on MS prevalence in Table 4? The numbers in the columns don’t add up to 100.
  • Figure 1 is with too figures and can potentially be merged to overlay social classes for easier comparison.

Author Response

Reviewer 2

This is a research paper aiming to understand the health disparity in metabolic syndrome by SES and gender among 42,146 working adults employed in the public administration, health care, and postal services in Spain. Overall, this is an interesting study that provides a great overview on how gender and SES affects metabolic syndrome at the different stage by gender among working adults. It would be nice if the study can be expanded to include all adults so that more women (for example housewives) can be included and accounted for. The manuscript is well-written with the following suggestions that need to be improved or addressed:

Conclusion in the abstract is repeating the results. Please conclude by summarizing overall study and impact in public health.

R: We modified the conclusion in the abstract as suggested by the reviewer: “From a public health perspective, SES could be strongly associated with the burden of MS; in an effort to reduce its prevalence, public health policies should focus on gender differences in socio-economic inequality and consider women with low socio-economic resources as a priority”.

The participant number is different from the number in the methods section.

R: We apologize for the mix-up. A total of 43,265 subjects decided to participate and signed the informed consent, however, of those, 42,146 presented available data on social class and were included in the analysis. We rectified the discrepancy in the methods section and modified the first line of the results section accordingly.

Can the authors please provide footnotes on how the statistics were done in each table in order for the audience to understand the comparison and tests being done?

R: According to the reviewer’s suggestion we added footnotes on the statistical tests employed for tables 1 and 2. In the case of tables 3, 4 and 5, crude prevalence ratios with 95% confidence intervals were calculated, as already indicated. 

Typo in line 185.

R: We believe the reviewer refers to the word normo-weight. We corrected the typo.

What were the numbers in the parentheses on MS prevalence in Table 4? The numbers in the columns don’t add up to 100.

R: The numbers in the parenthesis refer to the percentages of MS prevalence (by ATP-III or IDF) in each social class for men and women separately. The percentages do not sum up to 100% because they only show the percentage of subjects with MS in each social class; we are not showing the number of subjects without MS.

Figure 1 is with too figures and can potentially be merged to overlay social classes for easier comparison.

R: We thank the reviewer for the suggestion, however when we tried to overlap the three social classes, we noticed that the figure become too busy and thus difficult to understand.

Round 2

Reviewer 1 Report

Thank you for addressing my comments well. I have no further remarks.

This manuscript is a resubmission of an earlier submission. The following is a list of the peer review reports and author responses from that submission.